# Position effects of 22q13 rearrangements on candidate genes in Phelan-McDermid syndrome

**Sujata Srikanth**[1�], **Lavanya Jain**[2�], **Cinthya Zepeda-Mendoza**[3,4], **Lauren Cascio**[1], **Kelly Jones**[1], **Rini Pauly**[1], **Barb DuPont**[1], **Curtis Rogers**[1], **Sara Sarasua**[2], **Katy Phelan**[5], **Cynthia Morton**[3,4,6,7,8], **Luigi Boccuto**[1,9]*

1 Greenwood Genetic Center, Greenwood, SC, United States of America, 2 School of Nursing, Healthcare Genetics Program, Clemson University, Clemson, SC, United States of America, 3 Department of Obstetrics, Gynecology, and Reproductive Biology, Brigham and Women's Hospital, Boston, MA, United States of America, 4 Harvard Medical School, Boston, MA, United States of America, 5 Genetics Laboratory, Florida Cancer Specialists and Research Institute, Fort Myers, FL, United States of America, 6 Program in Medical and Population Genetics, Broad Institute of Harvard and MIT, Cambridge, MA, United States of America, 7 Department of Pathology, Brigham and Women's Hospital, Boston, MA, United States of America, 8 Division of Human Communication, Development and Hearing, School of Biological Sciences, Manchester Academic Health Science Center, Manchester, United Kingdom, 9 School of Nursing, College of Behavioral, Social and Health Sciences, Clemson University, Clemson, SC, United States of America

උ These authors contributed equally to this work.
* lboccut@clemson.edu

**Data Availability Statement:** All relevant data are within the manuscript and its Supporting Information files.

## Abstract

Phelan-McDermid syndrome (PMS) is a multi-system disorder characterized by significant variability in clinical presentation. The genetic etiology is also variable with differing sizes of deletions in the chromosome 22q13 region and types of genetic abnormalities (*e.g.*, terminal or interstitial deletions, translocations, ring chromosomes, or *SHANK3* variants). Position effects have been shown to affect gene expression and function and play a role in the clinical presentation of various genetic conditions. This study employed a topologically associating domain (TAD) analysis approach to investigate position effects of chromosomal rearrangements on selected candidate genes mapped to 22q13 in 81 individuals with PMS. Data collected were correlated with clinical information from these individuals and with expression and metabolic profiles of lymphoblastoid cells from selected cases. The data confirmed TAD predictions for genes encompassed in the deletions and the clinical and molecular data indicated clear differences among individuals with different 22q13 deletion sizes. The results of the study indicate a positive correlation between deletion size and phenotype severity in PMS and provide evidence of the contribution of other genes to the clinical variability in this developmental disorder by reduced gene expression and altered metabolomics.

## Introduction

Phelan-McDermid syndrome (PMS) is a genetic disorder caused by chromosomal rearrangements of the 22q13.3 region and pathogenic variants in *SHANK3* [1, 2]. This disorder presents with impressive genetic and clinical variability. The genetic alterations causing PMS are usually

**Funding:** This study was supported in part by the Greenwood Genetic Center Foundation and the Hope for 22q13 Gala. Additional support was provided by the Clemson University School of Nursing Center for Research on Health Disparities, the Clemson University Open Access Publishing Fund, and Clemson Support for Early Exploration and Development (CU SEED) Grant Program.

**Competing interests:** The authors have declared that no competing interests exist.

*de novo* and most cases present with terminal deletions, ranging from <100,000 bp to >9 Mb [3–5]. About 20% of the total PMS population shows deletions or duplications caused by unbalanced chromosomal translocations and in few rare cases the chromosomal imbalance is inherited from one parent with a balanced translocation [3, 5]. In other cases a ring chromosome 22 (r22) has been reported, either isolated or as part of a more complex rearrangement [6, 7].

By 2017, more than 1,500 individuals had registered with the Phelan-McDermid Syndrome Foundation [8]. About 1% of individuals with autism (1 in 8,000–15,000 including 22q13.3 deletions and *SHANK3* variants) are estimated to have PMS [9]. PMS features can appear from birth to childhood [9], prenatally [10] or even in late teen years [11]. Currently, there is no cure or treatment bespoke to specific needs of PMS, but therapies have been employed to manage individual symptoms. The early clinical presentation typically includes hypotonia and developmental delay accompanied by a weak cry and poor head control [9]. Individuals with PMS exhibit varying degrees of intellectual disability (ID), affecting particularly functional language, and variable rates of ASD ranging up to 94% [12–15]. About 40% of individuals with PMS develop seizures ranging from mild to severe. Gastrointestinal problems are commonly reported and include constipation, gastroesophageal reflux, and poor feeding. Individuals with PMS may have a high pain tolerance and tend to overheat due to poor regulation of body temperature and impaired sweating. Other recurrent features include minor dysmorphic traits such as large fleshy hands, dysplastic toenails, sacral dimples, and large prominent low-set ears. The severity of these clinical features ranges from severe to mild [9, 16].

Several reports have highlighted the importance of *SHANK3* haploinsufficiency or loss-of-function variants in the etiology of PMS [16, 17]. *SHANK3* plays an important role in synaptic function by encoding a scaffolding protein in the postsynaptic density of glutamatergic synapses [18]. However, individuals with larger deletions and therefore additional genes or regulatory regions involved, tend to be more severely affected, based largely on parental reports [4, 5]. A majority of individuals with PMS presenting with ASD and ID have *SHANK3* variants and deletions [19], while several other genes in the 22q13.3 region have been associated with different disorders and are therefore hypothesized to contribute to PMS-related symptoms [20, 21]. Some of the strongest evidence for the role of proximal genes of 22q13.3 in the phenotype of PMS comes from studies of interstitial deletions [22, 23]. One such study reports a girl-boy pair of twins with PMS that does not involve the *SHANK3* gene and this group of researchers found the clinical phenotype of patients without *SHANK3* deletion to be similar to that of patients with a *SHANK3* deletion. This confirms the hypothesis that 22q13 genes other than *SHANK3* have contributory roles towards PMS symptoms [24].

Metabolic imbalances have been known to contribute to clinical phenotypes in complex diseases in the past. One such study investigated neurodevelopmental manifestations in different disorders, namely, autism spectrum disorder (ASD), idiopathic-developmental delays, and Down syndrome (DS). The aim was to compare and correlate metabolic differences observed in these patients with respect to typically developed controls [25]. The rationale for this study was derived from other metabolomic impairments observed in patients with neurodevelopmental symptoms in the past, such as ASD and DS were associated with increased oxidative stress [26–28], and trisomy 21 resulting in DS was found to aid in the overexpression of amyloid protein leading to an increased risk of Alzheimer's Disease in DS patients [29]. These demonstrations point towards the possibility that changes in certain metabolic pathways could be responsible for severe neurodevelopmental features observed in PMS and it was this likelihood that was exploited as the premises for conducting metabolomic studies in our cohort.

Publicly available chromatin contact information has been used to annotate and predict regulatory elements and is increasingly valuable in correlating phenotypes with that of

neighboring genes (TAD or position effect analysis). TAD has been applied to PMS in the past to study the effect of disruption of genes resulting from breakpoints during the formation of r22 [7]. A transcription factor (*TCF20*) in a TAD with a gene involved in the immune system (*NFAM1*) was found to be misregulated due to the TAD perturbation caused by an inversion within the gene. This transcription factor is in turn associated with genes essential for neuro-development, leading to a plausible hypothesis that the TAD disruption contributed to the neurodevelopmental features observed in the 3-year old PMS patient. A similar rationale that was used for r22 formation for that publication was used in this project for patients with deletions. We applied a previously published strategy for the identification of position effect candidate genes [30] to a series of selected PMS subjects with chromosomal deletions. In our cohort, twelve position effect candidates were identified, and for nine of these genes we assessed a position effect impact on their expression and function in lymphoblastoid cell lines (LCLs). These genes were selected based on their roles in brain activity, expression in whole blood, and their location on the 22q13.3 genomic region. This study aimed to investigate the potential role of position effects on 22q13.3 genes and to assess the overall impact of such effects on the clinical, metabolic, and gene expression variability in PMS.

## Materials and methods

### Cohort of individuals with PMS to identify position effect candidate genes

A cohort of 114 individuals with PMS evaluated at the Greenwood Genetic Center was selected for this study to identify candidate position effect genes (Fig 1). All individuals carried a terminal deletion encompassing the 22q13.3 region; deletion breakpoints and sizes were assessed by microarray (a custom 4344K 60-mer oligo-array designed to interrogate 22q12.3-qter by Oxford Gene Technology (Oxford, UK) Sheet 1 in S1 File and converted from hg18 to hg19 using the UCSC LiftOver tool [31] Sheet 2 in S1 File. The custom 22q microarray spans approximately 319 genes between the regions 35–50 Mb on chromosome 22q with a resolution of 100 bp. In this technology, the sample from the patient, or family member, and a control sample are labeled differently with distinct fluorophores and are then co-hybridized to the micro-array. The ratio of the fluorescent intensity of the patient DNA to the control DNA is determined to measure the copy number present. Clinical features of these individuals with PMS have been reported previously [3] and were organized according to the International Classification of Diseases, 10th version [32] Sheet 3 in S1 File. The study was approved by the Self Regional Healthcare Institutional Review Board (IRB) for Human Research, under the number Pro00058564, IRB Study #80. Informed written consent was reviewed and signed by all participants and/or their legal guardians. Deidentified data were used for the computational prediction algorithm as discussed below.

### Selection of 12 candidate genes using position effect analysis

We used a computational prediction algorithm [30] to identify candidate position effect genes in the PMS cohort of 81 individuals with phenotypic information. In summary, the pipeline predicts the degree to which neighboring genes could be transcriptionally affected by a deletion and does so using a combination of genomic features such as inclusion in topological associating domains (TADs) and disruption of known or predicted regulatory contacts. Potential transcriptionally dysregulated genes are further evaluated by considering their haploinsufficiency (HI) scores as well as phenotypic overlap with the PMS individuals' clinical features. We searched for potential position effects by using windows extending up to 2 Mb on either side of the deletion breakpoints. Position effect candidates were ranked given their disruption of known or predicted regulatory contacts, location within the 2 Mb/TAD windows, HI scores

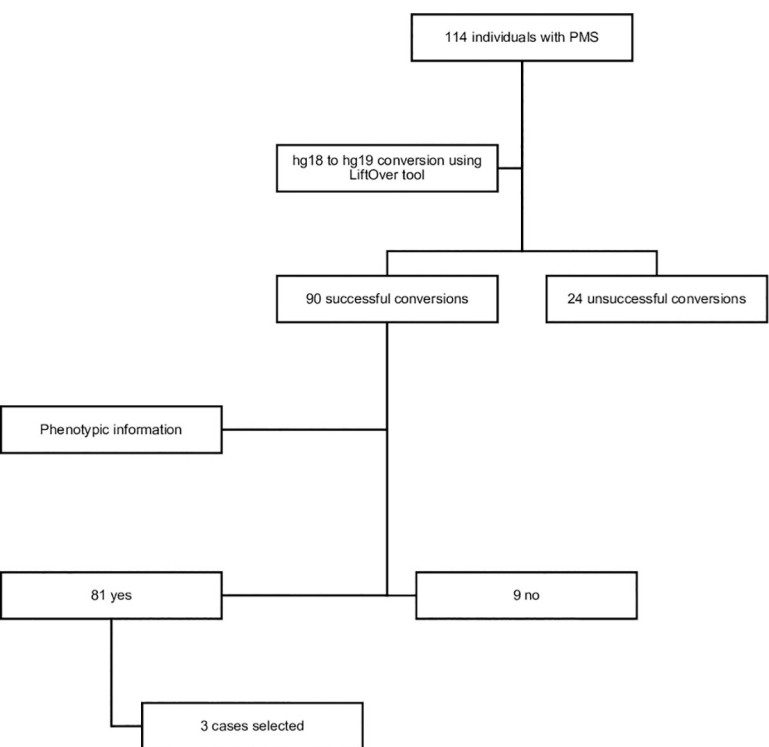

**Fig 1. Flowchart for the selection of individuals with PMS.** The initial cohort consisted of 114 patients; after applying a series of conditions mentioned above, 3 individuals were selected for this study. Two unrelated individuals were included in the study, reaching a total of 5 individuals.

(= <10%), and phenotypic overlap above the 75[th] percentile [30]. From this analysis, 12 candidate genes were identified. These 12 candidate genes were then selected for expression analysis as one method of validation.

## Validation sample

Five individuals with PMS (four males and one female, 3 from the cohort of 81 and two unrelated patients) were selected because of their diverse cytogenetic features for expression analysis and available lymphoblastoid cell lines (Table 1). Terminal deletions were confirmed independently by FISH and MLPA.

**Table 1. Selected individuals with PMS for expression and metabolic analysis.**

| Individual | Gender | Age at sample collection | Deletion start point (hg19) | Deletion end point (hg19) | Deletion size (Mb) | Definition of deletion size |
|---|---|---|---|---|---|---|
| PMS 1 | Male | 4 years | 42740931 | 51244566 | 8.503635 | Large |
| PMS 2 | Male | 12 years | 42816484 | 51244566 | 8.428082 | Large |
| PMS 3 | Male | 18 years | 43800990 | 51244566 | 7.443576 | Mid |
| PMS 4 | Female | 3 years | 47731071 | 51193680 | 3.462609 | Small |
| PMS 5 | Male | 18 years | 51123491 | 51224252 | 0.100761 | Small |

These individuals were classified into three groups, according to the size of their deletions: large- (PMS 1 and 2), mid- (PMS 3), and small-size (PMS 4 and 5). Such definition is arbitrary and purely based on the distribution of the genes in the 22q13 region. The sole purpose of the distinction of the deletion size into three categories is to provide a rough stratification of the contribution of different loci on 22q13. Position effects were assessed by gene expression, metabolic profiling, and clinical features.

## Lymphoblastoid Cell Lines (LCLs)

Peripheral blood samples were collected by venipuncture. Lymphoblastoid cell lines (LCLs) were obtained by immortalization with Epstein-Barr virus of lymphocytes isolated from the whole blood samples. The lymphoblastoid cell lines were cultured in Sigma RPMI-1640 with 15% fetal bovine serum (FBS) from Atlanta Biological (Flowery Branch, GA, USA) and 2 mM L-Glutamine, 100 U/ml Penicillin, and 100 µg/ml Streptomycin from Sigma-Aldrich (St. Louis, MO, USA).

## Gene expression using quantitative Real-time PCR (qPCR)

Quantitative Real-time PCR (qPCR) was used to measure gene expression for the 12 candidate position effect genes. Out of the 12 candidate genes, *FBLN1*, *SHANK3*, and *SCO2* did not show expression in blood cells. Hence, the remaining 9 genes were evaluated using qPCR (Fig 2). Five individuals with deletions in the 22q13 region (four males and one female) and two controls (one male and one female) were used for the study, with two biological replicates per sample. We selected a limited number of samples because we wanted to validate a mechanism, our aim was not to study the expression profile on any available patient with PMS. Hence, we identified 5 patients with different deletion sizes.

Total RNA was extracted from LCLs using TRIzol reagent according to the manufacturer's protocol. RNA concentration and quality were determined with a NanoDrop one Spectrophotometer (Thermo Fisher Scientific) and was confirmed to be free of genomic DNA contamination by PCR for Ribosomal protein L2 (*RPL2*), using primers spanning two exons and an intron. Five µg of RNA from each sample was subjected to reverse transcription using the iScript™ cDNA Synthesis Kit from Bio-Rad. qPCR was performed with the CFX96 Touch™

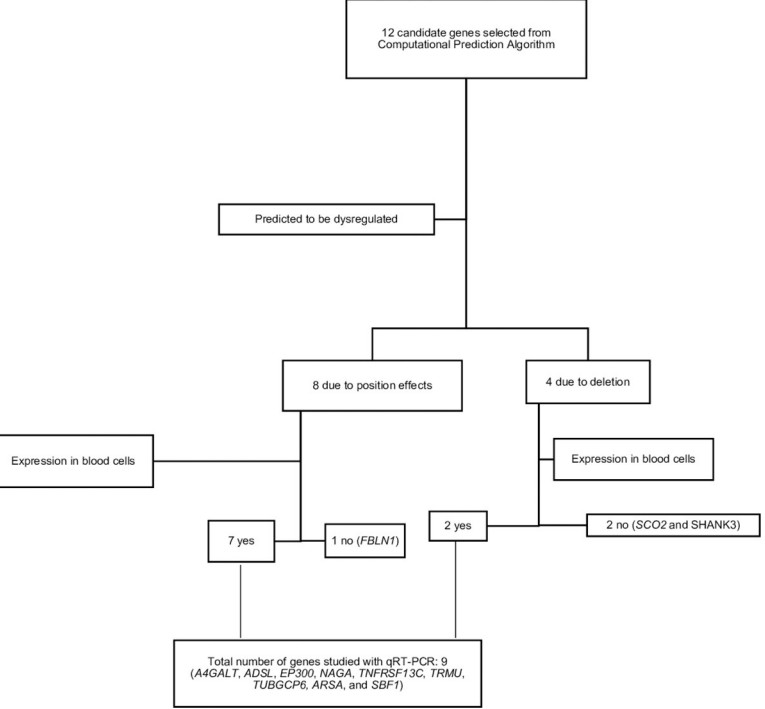

**Fig 2. Flowchart for selection of genes for qRT-PCR analysis.** Twelve candidate genes were identified with 9 genes being expressed in blood and able to be assessed with qRT-PCR. Seven of the 9 were predicted to be dysregulated due to position effects and 2 were predicted to be dysregulated due to deletion.

Real-Time PCR Detection System (Bio-Rad Laboratories). Samples were amplified in a 20 μl reaction containing 25 ng of cDNA from total RNA and 1 μl of gene-specific TaqMan probe-primer set. mRNA levels were normalized to a housekeeping gene, TATA-Box Binding Protein (TBP). Relative quantification for the gene of interest was calculated using the $2^{-\Delta\Delta Ct}$ calculation method in the software with respect to a control. Relative expression levels ≥1.5 were considered elevated and levels ≤0.66 were considered decreased.

## Metabolic profiling via Biolog metabolic arrays

Metabolic profiling was measured to assess the impact of position effects on metabolism. The Phenotype Mammalian MicroArray (PM-M) developed by Biolog (Hayward, CA, USA) is designed to assess the metabolic activity by measuring cellular production of NADH (nicotinamide adenine dinucleotide, reduced form) in the presence of different compounds. LCLs were used to measure metabolic dysregulation in Biolog Metabolic Arrays. These cell lines generated from the patient's blood sample via Epstein-Barr virus transfection were counted utilizing a TC20™ Automated Cell Counter in order to measure the amount and the percentage of viable cells. The methodology employs microplates with diverse molecules, which act either as energy sources (plates PM-M1 to M4) or as metabolic effectors (plates PM-M5 to M8). Each well contains a single chemical, and the production of NADH per well is monitored using a colorimetric redox dye chemistry. In other words, NADH production is used as an indicator of metabolic dysregulation. The energy sources include carbohydrates, nucleotides, carboxylic acids, and ketone bodies in plate PM-M1, and amino acids, both alone and as dipeptides in plates PM-M2 to M4. The metabolic effectors include ions (PM-M5), hormones, growth factors, and cytokines (PM-M6 to M8). The custom tryptophan plate (Trp) generated by Biolog in collaboration with the Greenwood Genetic Center was also employed in consideration of previously published data showing decreased utilization of tryptophan as an energy source by cells from individuals with Autism Spectrum Disorder (ASD) [33]. This plate consists of twelve 8-well columns containing glucose, empty well, tryptophan alone, and five dipeptides in which tryptophan is combined respectively with glycine, lysine, leucine, arginine, and alanine. PM-M plates were incubated with 20,000 lymphoblastoid cells per well (40,000/well for the Trp plate) in a volume of 50 μL, using the modified Biolog IF-M1 medium. Media for plates PM-M1 to M4 were prepared by adding the following to 100 mL of Biolog IF-M1: 1.1 mL of 100× penicillin/streptomycin solution, 0.16 mL of 200 mM Glutamine (final concentration 0.3 mM), and 5.3 mL of fetal bovine serum (final concentration 5%). For the Trp plate, 1.1 mL of fetal bovine serum was added instead of 5.3, for a final concentration of 1%. For plates PM-5 to M8, 5.5 mL of 100mM glucose (final concentration 5%) were added in place of the fetal bovine serum. Cells were incubated for 48 h at 37˚C in 5% $CO_2$. After this first incubation, Biolog Redox Dye Mix MB was added (10 μL/well) and plates were incubated under the same conditions for an additional 24 h, during which time cells metabolize the sole carbon source in the well. As the cells metabolize the carbon source, tetrazolium dye in the media is reduced, producing a purple color according to the amount of NADH generated. During the 24 h of exposure to dye, plates were incubated in the Omnilog system, which measured the optical density of each well every 15 minutes, generating 96 data points. The information collected during the 24 hours was analyzed by the kinetic software of the system to generate kinetic curves of the NADH generation for each well and calculate kinetic parameters, such as slope, endpoint, and area under the curve. At the end of the 24-h incubation, plates were analyzed utilizing a microplate reader with readings at 590 and 750 nm. The first value ($A_{590}$) indicated the highest absorbance peak of the redox dye and the second value ($A_{750}$) gave a measure of the background noise. The relative absorbance ($A_{590-750}$) was calculated per well.

For Phenotype Mammalian data, the absorbance endpoint readings and the 96 data points of kinetic optical density collected over the 24 h of incubation with the tetrazolium dye in the Omnilog system were used for data normalization and statistical analysis using R (opm R package) [34]. Readings were normalized using the triplicate absorbance readings from the corresponding empty plate (plates run with no cells, just media and dye). These values were then transformed to a logarithmic scale for the analysis and compared versus the average values generated by 50 LCLs from healthy controls, previously utilized in a study on ASD [33]. Our goal was to identify wells in which the levels of NADH generated by PMS cells were significantly different from ones measured in controls. We utilized the R package to implement the non-parametric Mann-Whitney approach of a two-sided *t*-test with the cut-off of *p*-value ≤0.05. Using this approach, metabolites differentially metabolized were identified distinguishing individuals with PMS and controls.

## Results

The computational prediction algorithm, which considered the clinical features and the 22q13 deletion breakpoints from 81 individuals with PMS, generated a list of 12 genes predicted to be dysregulated. As a validation, the expression levels of these genes were measured in five individuals. Metabolic profiling was performed to assess the response of LCLs from the five individuals to different compounds.

### Computational prediction algorithm

Twelve genes were predicted to be dysregulated: eight due to position effects and four due to deletion (Fig 4). The eight genes identified as potential candidates for transcriptional misregulation due to long-range position effects were *A4GALT*, *ADSL*, *EP300*, *FBLN1*, *NAGA*, *TNFRSF13C*, *TRMU*, and *TUBGCP6* (Sheet 4 in S1 File). Except for *A4GALT*, all candidate genes have been associated with phenotypic traits compatible with the clinical features reported in the cases analyzed (Sheet 3 in S1 File). Only *ADSL* and *EP300* have haploinsufficiency (HI) scores = <10%, which makes them even more meaningful candidates for functional follow-up for these cases. Noteworthy, not all candidate genes were detected in every individual with PMS analyzed. Some of the genes were detected in one individual (*i.e.*, *TUBGCP6*) and others in up to nine individuals (*i.e.*, *ADSL*), which indicates a heterogeneous contribution of position effects to the overall clinical presentation.

The four candidate genes identified for transcriptional misregulation due to short-range effects of the deletion were *ARSA*, *SBF1*, *SCO2*, and *SHANK3*. Contrary to the position effect candidates which could act over long genomic distances, these genes are almost invariably included within the deletion coordinates (Fig 3A and Table 2). Of note, *SHANK3* was deleted in all cases analyzed. The functions of all of these genes are described in the Table in S2 File. The presence or absence of these selected genes and genes in the TAD region for each patient are reported in Table 2. Fig 4 shows exactly the same data displayed with two different shading schemes according to the genes predicted to be dysregulated by long-range or short-range effects.

### Clinical features of selected individuals with PMS

Clinical features are summarized in Table 3 with details following. Clinical traits for patients 2 and 5 were described in detail in a previous study [35] and only the most relevant features are reported herein.

**PMS 1.**　PMS 1 is a 14-year-old Caucasian male, born from non-consanguineous parents after an uneventful pregnancy. He was diagnosed with PMS secondary to an 8.5 Mb *de novo* terminal deletion with the proximal breakpoint mapped to 22q13.2.

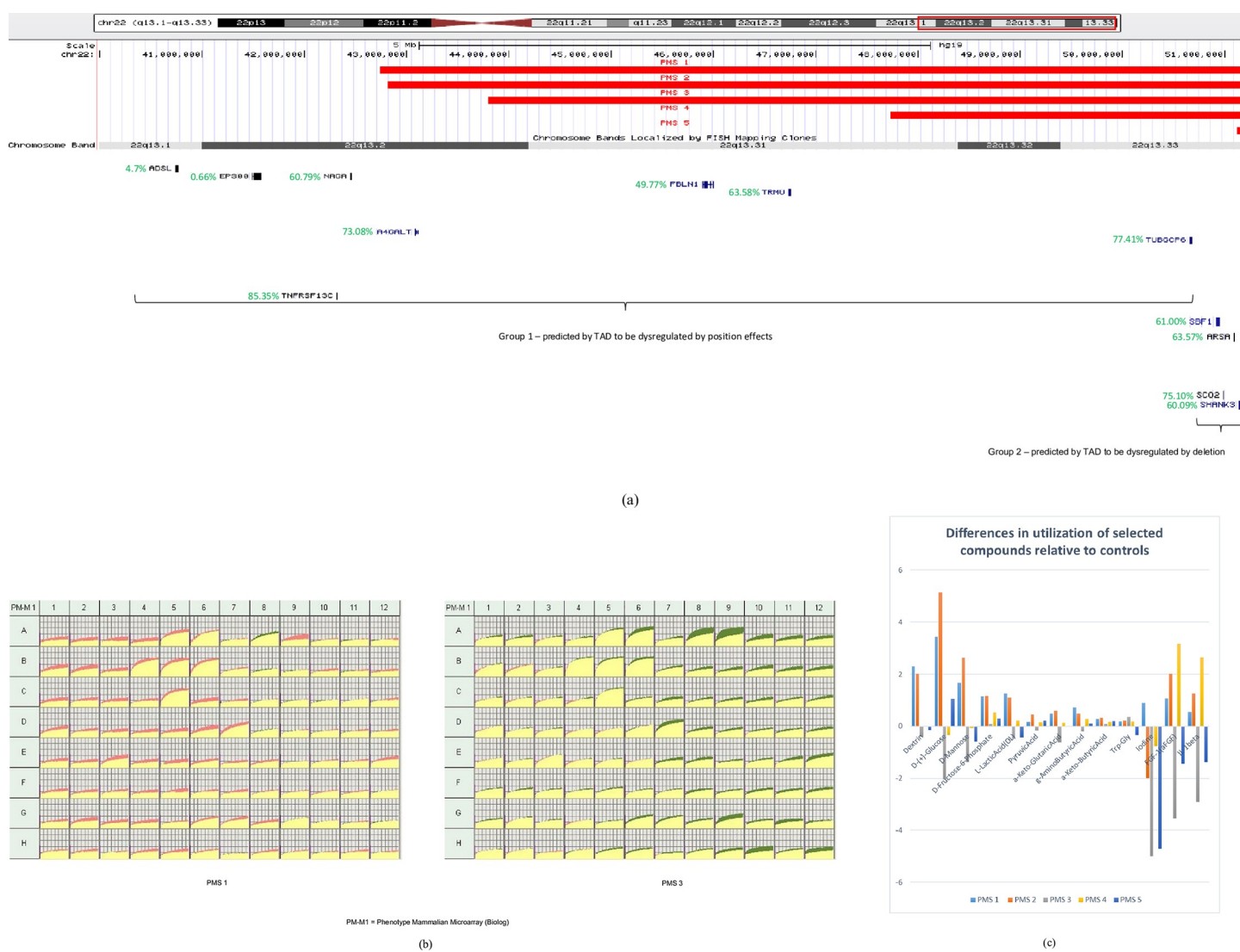

**Fig 3. (a) UCSC Genome Browser (GRCh37/hg19 build) showing deletion breakpoints for each of the five chosen individuals with TAD region (2 Mb upstream from the deletion starting breakpoint) in gray dotted line, and genes predicted to be dysregulated by position effects or deletion at the bottom.** The genes shown in group 1 were predicted to be dysregulated by position effect. The genes shown in group 2 were predicted to be affected when deleted. Haploinsufficiency scores (%HI) from DECIPHER v9.30 are provided in green. High ranks (*e.g.*, 0–10%) indicate a gene is more likely to exhibit haploinsufficiency. **(b) Parametric analyses of PM-M1 plate for PMS 1 (left) and PMS 3 (right).** These analyses were generated by OD (optical density) measurement every 15 minutes for 24 hours after adding the dye. Each smaller box represents each well. The yellow area represents the overlap between the control and the sample, pink represents the underutilization of the substrate by the sample and green represents the overutilization of the substrate by the sample. **(c) Bar chart representing an increase or decrease in utilization of selected compounds on Biolog plates for each individual with PMS.** The *x*-axis represents selected compounds on Biolog plates (PM-M1 through PM-M8 and PM-Trp). The *y*-axis represents the difference between the utilization for individuals and the average of 50 controls. The line at 0 represents the utilization for the average of 50 controls.

At the evaluation, PMS 1 was non-ambulatory but would take steps with maximum support. He was not using a walker. His mother reported that he had regression in his development when he had any type of illness. He continued to receive physical and occupational therapy.

His interim history revealed that he had hip surgery at the age of 6 years and had significant complications requiring hospitalization with the removal of plates 2.5–3 years later. His parents reported respiratory issues and problems with the regulation of body temperature. He has also been hospitalized for kidney reflux and flu and fevers of unknown etiology. His

**Table 2. Expression levels of 9 genes on 22q13 investigated as candidates for transcriptional misregulation.**

| Gene | Gene coordinates, hg19 | Patients and deletion sizes | | | | | Average 2^{-ΔΔCt} when not deleted | Average 2^{-ΔΔCt} when deleted | Average 2^{-ΔΔCt} when within 2 Mb of deletion region |
|---|---|---|---|---|---|---|---|---|---|
| | | PMS 1 8.5 Mb | PMS 2 8.4 Mb | PMS 3 7.4 Mb | PMS 4 3.5 Mb | PMS 5 0.1 Mb | | | |
| ADSL | chr22:40,742,504–40,762,575 | ✓ | ✓ | ✓ | ✓ | ✓ | 0.92 | N/A | 1.03 |
| EP300 | chr22:41,488,614–41,576,081 | ✓ | ✓ | ✓ | ✓ | ✓ | 0.88 | N/A | 0.85 |
| TNFRSF13C | chr22:42,321,036–42,322,821 | ✓ | ✓ | ✓ | ✓ | ✓ | 0.82 | N/A | 0.79 |
| NAGA | chr22:42,454,338–42,466,846 | ✓ | ✓ | ✓ | ✓ | ✓ | 0.81 | N/A | 0.99 |
| A4GALT | chr22:43,088,127–43,116,876 | ✗ | ✗ | ✓ | ✓ | ✓ | 2.71 | 1.49 | 2.23 |
| TRMU | chr22:46,731,298–46,753,237 | ✗ | ✗ | ✗ | ✓ | ✓ | 0.9 | 0.51 | 0.87 |
| TUBGCP6 | chr22:50,656,118–50,683,400 | ✗ | ✗ | ✗ | ✗ | ✓ | 1.1 | 0.6 | 1.1 |
| SBF1 | chr22:50,883,431–50,913,464 | ✗ | ✗ | ✗ | ✗ | ✓ | 1.19 | 0.53 | 1.19 |
| ARSA | chr22:51,061,182–51,066,601 | ✗ | ✗ | ✗ | ✗ | ✓ | 1.15 | 0.85 | 1.15 |

The '✓' represents presence of the gene in the individual, the '✗' represents deletion of the gene in the individual. Shaded cells represent the genes within 2 Mb for the individual mentioned in the first row of that column. Hg19 UCSC genome browser [31].

**DELETED GENES IN SHADED CELLS**

| Genes | | ADSL | EP300 | TNFRSF13C | NAGA | A4GALT | TRMU | TUBGCP6 | SBF1 | ARSA |
|---|---|---|---|---|---|---|---|---|---|---|
| Distance from chromosome end: | | 10.5 Mb | 9.6 Mb | 8.9 Mb | 8.8 Mb | 8.1 Mb | 4.5 Mb | 601 kb | 374 kb | 194 kb |
| Haploinsufficiency score (%HI): | | 4.7 | 0.66 | 85.35 | 60.79 | 73.08 | 63.58 | 77.41 | 61 | 63.57 |
| Predicted to be dysregulated by: | | Position | Position | Position | Position | Position | Position | Position | Deletion | Deletion |
| | Deletion size | | | | | | | | | |
| PMS 1 | 8.5 Mb | 1.03 | 0.87 | 0.82 | 0.66 | 1.52 | 0.56 | 0.54 | 0.51 | 0.73 |
| PMS 2 | 8.4 Mb | 0.97 | 0.83 | 0.83 | 1.54 | 1.46 | 0.45 | 0.62 | 0.53 | 0.84 |
| PMS 3 | 7.4 Mb | 1 | 0.9 | 0.66 | 0.78 | 2.23 | 0.54 | 0.69 | 0.53 | 0.6 |
| PMS 4 | 3.5 Mb | 0.64 | 0.83 | 0.83 | 0.55 | 2.84 | 0.87 | 0.56 | 0.55 | 1.42 |
| PMS 5 | 0.1 Mb | 1.02 | 1 | 0.84 | 0.79 | 3.14 | 0.94 | 1.1 | 1.19 | 1.15 |

>0.5 and ≤0.66 ⬛ ≥1.5 and ≤2 ⬛ >2 and ≤3 ⬛ >3 ⬛ Expression when deleted

(a)

**NON-DELETED (PRESERVED) GENES IN SHADED CELLS**

| Genes | | ADSL | EP300 | TNFRSF13C | NAGA | A4GALT | TRMU | TUBGCP6 | SBF1 | ARSA |
|---|---|---|---|---|---|---|---|---|---|---|
| Distance from chromosome end: | | 10.5 Mb | 9.6 Mb | 8.9 Mb | 8.8 Mb | 8.1 Mb | 4.5 Mb | 601 kb | 374 kb | 194 kb |
| Haploinsufficiency score (%HI): | | 4.7 | 0.66 | 85.35 | 60.79 | 73.08 | 63.58 | 77.41 | 61 | 63.57 |
| Predicted to be dysregulated by: | | Position | Position | Position | Position | Position | Position | Position | Deletion | Deletion |
| | Deletion size | | | | | | | | | |
| PMS 1 | 8.5 Mb | 1.03 | 0.87 | 0.82 | 0.66 | 1.52 | 0.56 | 0.54 | 0.51 | 0.73 |
| PMS 2 | 8.4 Mb | 0.97 | 0.83 | 0.83 | 1.54 | 1.46 | 0.45 | 0.62 | 0.53 | 0.84 |
| PMS 3 | 7.4 Mb | 1 | 0.9 | 0.66 | 0.78 | 2.23 | 0.54 | 0.69 | 0.53 | 0.6 |
| PMS 4 | 3.5 Mb | 0.64 | 0.83 | 0.83 | 0.55 | 2.84 | 0.87 | 0.56 | 0.55 | 1.42 |
| PMS 5 | 0.1 Mb | 1.02 | 1 | 0.84 | 0.79 | 3.14 | 0.94 | 1.1 | 1.19 | 1.15 |

≤0.5 ⬛ >0.5 and ≤0.66 ⬛ ≥1.5 and ≤2 ⬛ Expression when not deleted (preserved)

(b)

**Fig 4.** (a) Heat map of expression of genes in five individuals relative to control with deleted genes in gray cells. (b) Heat map of expression of genes in the five individuals relative to control with pre genes in gray cells. Numbers are relative fold changes in expression (2-ΔΔCt). These heat maps give an account of exactly the same data displayed with two different shading schemes, expression differences due to dysregulation by position effects in (a) and by deletion in (b). The third row in both show causes of dysregulation predicted for each gene by position effect analysis as either dysregulated by position effect or by deletion.

**Table 3. Summary of clinical features for each individual with PMS.**

| Signs and symptoms | PMS 1 | PMS 2 | PMS 3 | PMS 4 | PMS 5 |
|---|---|---|---|---|---|
| Developmental delay | ✓ | ✓ | ✓ | ✓ | ✓ |
| Language delay | ✓ | ✓ | ✓ | ✓ | ✓ |
| Seizures | ✓ | ✓ | | | ✓ |
| Hypotonia | ✓ | ✓ | ✓ | | ✓ |
| Sleep issues | ✓ | ✓ | | | ✓ |
| Gastrointestinal issues (including liver steatosis) | ✓ | ✓ | | | ✓ |
| *Cranial dysmorphic traits* | | | | | |
| • Abnormal head shape/facial asymmetry | ✓ | | ✓ | | ✓ |
| • Periorbital puffiness | ✓ | | ✓ | | |
| • Eye abnormalities | ✓ | ✓ | ✓ | | |
| • Ear abnormalities | ✓ | ✓ | ✓ | | ✓ |
| • Broad nasal tip/bridge | ✓ | ✓ | ✓ | | ✓ |
| Behavioral issues | ✓ | ✓ | ✓ | ✓ | ✓ |
| Kidney reflux | ✓ | | ✓ | | |
| Skeletal abnormalities | ✓ | ✓ | | | |
| Hyperconvex fingernails | ✓ | | ✓ | | |

current medications included Risperidone 1.5 mg twice a day, Ativan gel as needed, MiraLAX (Polyethylene glycol 3350) as needed, and Tylenol (Paracetamol) or Motrin (Ibuprofen) as needed. He frequently requires ear tubes due to recurrent otitis media and had had six sets at the time of evaluation. Seizures were reported but no current medication was prescribed. His swallowing had improved but he was having problems with constipation.

He had a height of roughly 130 cm (~5th percentile). His weight was 78 pounds (50th-75th percentile), and head circumference was 56.8 cm (slightly above the 97th percentile). He was a healthy-appearing 13-year-old male with a somewhat round face with mild facial asymmetry and plagiocephaly with flattening of the right occiput and the left forehead. The eyes revealed a bilateral red reflex. The nasal tip was broad, and he had mild periorbital puffiness. The ears were slightly asymmetric. Two small sacral dimples were noted. He had some mild stiffness of his extremities, but they could be gradually fully extended. He had mild hyperreflexia at the knees and mild ankle clonus. The fingernails were slightly hyperconvex. The toenails were thin and flaky with some hyperplasia of individual nails. His seizures were reported to be occurring more frequently. He had had a normal EEG, but the parents had noted several tonic clonic seizures.

**PMS 2.** PMS 2 is a 26-year-old Hispanic-Caucasian male. No parental consanguinity was reported. Prenatal chromosome analysis was performed due to advanced maternal age and revealed a 22q13.2-q13.33 deletion, later confirmed by microarray as an 8.4 Mb *de novo* terminal deletion, validating the clinical diagnosis of PMS.

He was born at 32 weeks of a pregnancy complicated by HELLP syndrome (Hemolysis, Elevated Liver enzymes, and Low Platelet count) and preeclampsia. Birth length was 33 cm (<3rd centile, 50th centile for 24.5 weeks of gestation) and weight was 1.1 kg (<3rd centile, 50th centile for 27.5 weeks of gestation). The early developmental history was characterized by feeding issues that required a feeding tube at age 4 years, severe delay of all early milestones, and prescription glasses for extreme nearsightedness at 4 years of age. He started to walk with some assistance at age 8. A brain MRI detected an arachnoid cyst around age 10. Seizures began at 14 and became constantly more frequent, requiring daily treatment with ONFI (Clobazam), Keppra (Levetiracetam), and Banzel (rufinamide). Sleeping issues were reported in association

with the beginning of the therapy with anti-epileptic drugs and seemed to worsen gastrointestinal issues. At age 22 his height was 157.4 cm (<3rd centile, 50th centile for age 13) and his weight was 36.3 kg (<3rd centile, 50th centile for age 11). The patient had jaw surgery, spinal surgery due to scoliosis, and hip surgery due to hip dysplasia. He developed moderate steatosis and a cataract. After back surgery, he lost the ability to walk alone or crawl but can walk with assistance.

**PMS 3.** PMS 3 is a 20-year-old Caucasian male with a diagnosis of PMS secondary to the detection by array-CGH of a 7.4 Mb terminal 22q13 deletion. He had a history of vertebral anomalies, ear tags, epibulbar dermoids, and hemifacial microsomia, suggesting a diagnosis of facioauriculovertebral dysplasia spectrum or Goldenhar syndrome. Significant developmental delay was reported followed by the diagnosis of the 22q13.3 deletion, along with fairly continuous drooling, partially managed with Scopolamine patches. He suffered a spiral fracture of his right tibia and has a history of kidney reflux which had resolved.

At the evaluation, his head circumference was 54.5 cm (35th centile). He presented with mild facial asymmetry due to the left side of the face being slightly smaller than the right. His face had a somewhat triangular shape with mild prominence of his chin. He had a history of mild periorbital puffiness, which had resolved. His habitus appeared somewhat tall and thin. His eyelashes were very long and thick. He had a broad nasal tip, remnants of ear tags present on the left side of his face, mildly prominent ears, thin and hyperconvex fingernails, and generalized hypotonia.

**PMS 4.** PMS 4 is a 7-year-old Caucasian female with a 22q13 terminal deletion of 3.5 Mb detected by array-CGH. She had dyspraxia and verbal apraxia. She used to form various syllables and make animal sounds but had regressed. She was very interactive, had good play skills, and was very aware of her environment. She showed high levels of anxiety and had problems with adaptive behavior.

At birth, her weight was 2.67 kg (10th-25th centile), height was 49 cm (50th centile), and head circumference was 33 cm (25th centile). She suffered from jaundice. At the evaluation, at age 5 years, her weight was 12 kg and her height was 93 cm (both <3rd centile, as well as her head circumference); her mother reported some progress in speech, leading to speaking a few words and phrases, but mostly non-functional, and singing some songs.

**PMS 5.** PMS 5 is a 21-year-old Caucasian male carrying a 100.76 Kb 22q13 deletion encompassing the *SHANK3* gene detected by array-CGH. He was born from non-consanguineous parents at the 38th week of a pregnancy complicated by hyperemesis gravidarum and threatened miscarriage. Length at birth was 52 cm (>90th centile), weight was 3,650 gr (75th-90th centile), head circumference was 36 cm (>90th centile), and Apgar scores were 7 and 9. Digestive problems and hypotonia were reported in the first years of life. Mild delay of developmental milestones was recorded: babbling at 9–10 months, crawling at 13 months, and independent walking at 18 months. During a clinical evaluation at age 4 years, following an epileptic episode, ID and disharmonic organization of personality were diagnosed, significant sleep disturbance was also noted, and diffuse hypomyelination of the brain's white matter was detected by an MRI. Seizures became more frequent over time and eventually the continuous and severe episodes of drug-resistant seizures Lennox Gastaut-type required a partial corpus callosotomy, which provided limited benefit. The diagnosis of PMS was confirmed by CGH microarray (hg19), which detected two independent chromosomal rearrangements: a 12p13.2 trisomy (10,572,751–10,593,748) and a 22q13.33 (51,123,491–51,224,252) deletion, including *SHANK3*; parental tests confirmed both rearrangements as *de novo*. Since there is no evidence in the literature supporting a pathogenic role for the 12p13.2 trisomy, we included the patient in the study assuming that the phenotype was entirely determined by the 22q13.33 deletion.

At the evaluation, PMS 5's height was around the 50th centile, weight between 50th and 75th centile, and head circumference between 75th and 90th centile. He presented with

dolichocephaly, broad nasal bridge, low-set ears, and protruding lower lip. He was nonverbal, hyperactive, slightly irritable, had reports of constipation and reduced pain perception, and was diagnosed with moderate liver steatosis.

### Expression analysis by qPCR for selected candidate genes

TAD computational analysis predicted 12 genes mapping in 22q13 to have abnormal expression. RNA levels of the 9 of these 12 genes expressed in blood were measured in LCLs via qPCR.

Deletion sizes for each patient are shown in Fig 3A. A consistent pattern was observed according to whether the gene was present or deleted in the patient, with the deleted genes generally showing overall decreased expression as compared to the preserved genes (Figs 3A, 4A and Table 2). The enzyme *A4GALT* was elevated for all individuals with PMS, whether one copy was deleted or not.

PMS 1 (8.5 Mb). *A4GALT* had unexpectedly elevated expression in both individuals of this group (1.52 and 1.46). The remaining genes with haploid copy number had reduced expression in PMS 1, while preserved genes had expression levels within normal ranges.

PMS 2 (8.4 Mb). Reduced expression was observed for all of the six deleted genes. Among preserved genes, *NAGA* expression was increased by 50% (1.54) while the other genes had expression levels within normal ranges.

PMS 3 (7.4 Mb). *TNFRSF13C* had reduced expression (0.66) consistent with the TAD prediction. *A4GALT* had unexpectedly high expression (2.23) even though it was not deleted.

PMS 4 (3.5 Mb). Two of the four deleted genes showed reduced expression (*TUBGCP6* (0.56) and *SBF1* (0.55)). *A4GALT* transcript levels were elevated (2.84), as in other cases. *ADSL* (0.64) and *NAGA* (0.55) showed decreased expression even if not deleted.

PMS 5 (0.1 Mb). None of the genes tested for expression by qPCR are deleted in this individual. *A4GALT* mRNA levels were increased (3.14) with the remaining genes unaffected.

**Genes predicted to be dysregulated due to deletion (*ARSA*, *SBF1*).** *SBF1* showed a considerable reduction in expression when deleted as compared to when preserved. *ARSA* was deemed to have inconclusive results due to variable expression.

**Genes predicted to be dysregulated due to position effects (*A4GALT*, *ADSL*, *EP300*, *NAGA*, *TNFRSF13C*, *TRMU*, and *TUBGCP6*).** *ADSL* showed reduced expression only in PMS 4 although this gene was not included in the 2Mb TAD region for this individual. This could be due to some other mechanism. Whereas similarly reduced expression is seen in PMS 3 for *TNFRSF13C*, where unlike PMS 4, this is attributed to position effect due to this gene being within the ±2 Mb TAD region from the deletion for this individual. *EP300* was preserved in all individuals and showed no changes pertaining to position effects. *NAGA* was not deleted in any of the five individuals; however, it showed an increase in expression in PMS 2 as well as a decrease in PMS 1 and PMS 4. *A4GALT* showed an increased expression overall (even when deleted in PMS 1 and PMS 2) but shows an especially higher expression when preserved (PMS 4 and PMS 5), while being considerably higher when in the TAD region for PMS3. *TRMU*, when present in the TAD region (PMS 4), shows increased expression as compared to when deleted, but a lower expression than when preserved (PMS 5), pointing towards a diminished position effect. *TUBGCP6* had decreased expression when deleted but was unaffected when within the TAD region.

### Data processing and statistical analysis of Phenotype Mammalian MicroArray (PM-M) data

Data generated by the metabolic arrays indicated a common trend in 4 out of 5 individuals, with PMS 5 (the individual with the smallest deletion limited to a major part of *SHANK3*)

showing a different profile (Fig 2 in S2 File). The utilization of energy sources (PM-M1 to M4 plates) was largely decreased in the two individuals with large deletions (PMS 1 and 2), which is consistent with a more severe metabolic imbalance due to a larger number of genes deleted. The findings from the PM-M1 plate showed similar profiles in the samples with small- and mid-size deletions, with a dominating trend toward increased utilization of carbon energy sources, although PMS 5 produced significant differences (adjusted $p$ value <0.05) only in 45 out of the 96 wells of the array, versus 70 and 83 in PMS 4 and 3, respectively (Fig 2 in S2 File). Therefore, the abnormal utilization of energy sources from the PM-M1 plate seems to correlate with the deletion size: PMS 5 has the lowest number of wells with significant differences while PMS 1 and 2 have the highest ones.

Interestingly, PMS 3 alone showed increased NADH levels in the presence of the intermediates of the Krebs cycle (Fig 2 in S2 File), this might suggest a different type of energetic disruption as compared to the other samples.

In the plates containing amino acids and dipeptides as energy sources (PM-M2 to M4 and Trp), a similar trend as for PM-M1 was observed: PMS 1 and 2 produced lower levels of NADH than controls in most wells, while PMS 3 and 5 produced mostly higher levels. PMS 4, on the other hand, produced a metabolic profile for amino acids more similar to the cell lines with large deletions (prevalence of lower NADH levels) than to the ones with similar deletion sizes (PMS 3 and 5). Reduced utilization of tryptophan as an energy source has been reported in individuals with ASD [33], and the same trend has been observed in PMS 2, 3, and 4, while in PMS 1 the differences did not reach statistical significance and in PMS 5 an increased utilization of tryptophan as compared to controls was detected (Fig 2 in S2 File).

In plates PM-M5 to M8 the energy source (glucose) was distributed equally to all cells via suspension media and the goal was to assess how the cells metabolized the provided glucose in the presence of different metabolic effectors: ions (PM-M5), growth factors (PM-M7), cytokines (PM-M7 and M8), and other hormones (PM-M6 to M8).

The metabolic trend emerging from these arrays confirmed that the two cell lines with the largest deletions (PMS 1 and 2) produced significant metabolic disruption as compared to controls ($p$ values <0.05, Sheet 6 in S1 File), although PMS 1 showed significant differences only in 43/96 (44.8%) wells from plate PM-M6 and 15/96 (15.6%) from PM-M7, containing growth factors, cytokines, and other hormones. On the other hand, PMS 3 and 5 showed an increased production of NADH ($p$ <0.05) across all wells from plates PM-M5 to M8, as observed also for plates PM-M1 to M4. The metabolic profile of PMS 4 was in agreement with the one noted in plates PM-M2 to M4: reduced energetic response to most wells, precisely 76/96 (79.2%) from PM-M5, 88/96 (91.7%) from PM-M6, 94/96 (97.9%) from PM-M7 and 96/96 (100%) from PM-M8 (Sheet 6 in S1 File).

Only three compounds shared similar trends across the five cell lines: manganese chloride (PM-M5 well C8, increased NADH production), iodine (PM-M5, wells C10-C11, increased NADH levels in the four samples in which it was significant), and D-glucose-1-phosphate (PM-M1, well B2, decreased NADH levels in the four samples in which it was significant).

## Discussion

Many studies have generated consistent evidence suggesting a prominent role for *SHANK3* haploinsufficiency or loss-of-function variants in the pathogenesis of the neurobehavioral features of PMS [16, 17]. However, the extensive clinical variability exhibited by individuals with PMS and the different sizes of chromosomal rearrangements detected in the 22q13 region have led to the hypothesis that other genes may contribute to PMS phenotypes. Some reports have indeed demonstrated a correlation between 22q13 deletion sizes and the severity of

certain clinical traits, such as developmental and language delay [4, 5], and other studies have proposed a series of genes mapping in 22q13 whose haploinsufficiency or disruption will likely determine alterations leading to signs and symptoms compatible with PMS phenotypes [20, 21]. In order to investigate further pathogenic mechanisms in PMS, we explored the position effect of genes in 22q13 by TAD analysis of a cohort of 81 individuals with PMS. The results of the predictions were validated by assessing clinical information, expression studies, and metabolomic arrays in five selected cases.

Genes that were predicted to be dysregulated due to deletion seem to follow a general trend of lower expression upon deletion, particularly for *SBF1*. Among candidate genes for dysregulation due to position effects, *ADSL* and *EP300* do not show significantly distinct expression values when in the 2 Mb TAD region as compared to when farther away. However, in PMS 3, *TNFRSF13C* shows a slightly lower expression as compared to when not in the TAD region. *A4GALT* represents an exception to some extent because its expression pattern was not as expected and differs from ones of surrounding genes with increased expression regardless of the deletion size. One possible explanation for this unexpected trend is that it encodes an enzyme and could therefore be regulated by a compensatory mechanism aimed to prevent haploinsufficiency: in case one allele was either lost by deletion or inactivated, the remaining allele would be over-expressed to maintain a physiological level of protein. This result implies that in a contiguous gene deletion syndrome even deleted genes can have a presumably compensatory increase in expression. Both *TRMU* and *TUBGCP6*, when in the TAD region, show expression levels close to that of the control.

*NAGA* in PMS 2 was predicted to be in the TAD region for a possible position effect—it shows an unexpected increase in expression pointing towards potential position effects from the deleted region of chromatin. One caveat for all the five cases considered in this paper is that there may be structural complexity at the breakpoints of the deletions because those regions were not sequenced. For this project's analysis, the minimum size of deletions was determined by the first deleted probe in the microarray. In PMS 5, all genes in the TAD show increased expression as compared to control, pointing towards a probable position effect mediated by the deletion of the major part of SHANK3. All these results hold true with the hypothesis that rearrangements in the 22q13 regions cause position effects among candidate genes within the same region. According to %HI (shown in Fig 3A), *EP300*, and *ADSL* are most susceptible to be haploinsufficient. *ADSL* deficiency is known to cause epilepsy or autistic features among other characteristics, and *EP300* is known to play a role in Rubinstein-Taybi syndrome which includes features overlapping with PMS (*e.g.*, typical facial features, microcephaly, and ID) (S2 File). In PMS 4, *ADSL* shows an unexpected decrease in expression, which could be possibly attributed to haploinsufficiency. In order to dissect how these effects manifest in PMS individuals, clinical and metabolic information was correlated with this expression data. For the expression data, as shown in Fig 1C and 1D, the values ≥1.5 have been considered increased with respect to the control, and values ≤0.66 have been considered decreased with respect to the control.

Three genes, namely, *SHANK3*, *SCO2*, and *FBLN1*, did not show any expression in blood cells, resulting in exclusion from the qPCR studies. Out of these, *FBLN1* was predicted to be dysregulated due to a position effect, whereas *SHANK3* and *SCO2* were predicted to be dysregulated when deleted. *SHANK3* was deleted in all individuals. *FBLN1* was deleted in PMS 1 through PMS 3, and *SCO2* was deleted in PMS 1 through PMS 4. As *SHANK3* is the primary gene correlated with PMS, all individuals show symptoms that are characterized as PMS symptoms. *SCO2* is responsible for causing Fatal Infantile Cardioencephalomyopathy, and even though this disorder is not seen in individuals with PMS, this gene might be responsible indirectly for the heart-related defects observed.

## PMS 1

This male individual shows clinical features that align with the genetic and metabolic features. For example, *TRMU* was deleted and had decreased gene expression. This gene is associated with deafness (S2 File) and this individual has had several placements of ear tubes due to recurrent otitis media. *TUBGCP6* was deleted and had decreased expression. Its absence can cause delayed psychomotor development and visual impairment (S2 File). PMS 1 presents with developmental regression and ocular bilateral red reflex that was observed whenever the patient was ill. *SBF1* is associated with Charcot-Marie-Tooth disease 4B3 and its deletion in this individual may be the cause of mild stiffness of the extremities, mild hyperreflexia, and mild ankle clonus. *SHANK3* is associated with seizures and other dysmorphic features seen in PMS 1 (S2 File). General decreased utilization for most metabolites on all Biolog plates was observed (*i.e.*, carbon energy sources, amino acids and dipeptides as energy sources, and glucose utilization in the presence of different metabolic effectors–ions, growth factors, cytokines, and other hormones); only 44.8% and 15.6% significant differences were seen for plates PM-M6 and M7, respectively.

## PMS 2

Severe delay of all early milestones was noted in this male individual with deletion of *A4GALT*, associated with general cognitive ability, even though *A4GALT* expression was increased overall for all individuals. *TUBGCP6* showed reduced expression consistent with its deletion. *TRMU*, also deleted in this individual, has been related to transient infantile liver failure (S2 File). PMS 2 was born with HELLP syndrome (hemolysis, elevated liver enzymes, and low platelet count) and also shows increased liver echogenicity and of the liver/kidney and liver/vascular gradient, as observed in moderate steatosis. He also was prescribed glasses for extreme nearsightedness which could be a manifestation of *TUBGCP6* deletion. *SHANK3* deletion is a reasonable etiology for seizures and other dysmorphic features (S2 File). *SBF1* and *ARSA* are also deleted in this individual. The expression for *NAGA* is seen to be increased considerably, augmenting its expression because it is within a distance of ±2 Mb from the deletion. All the Biolog plates, including Tryptophan, show decreased utilization for energy sources as compared to controls.

## PMS 3

Significant developmental delay in this individual could be a result of *TUBGCP6* deletion. *SBF1* deletion in PMS 3 could underlie the generalized hypotonia. Deletions in *SHANK3* are associated with dysmorphic features, which are observed in this individual (S2 File). The Biolog profile is similar to that of individuals with small deletions, *i.e.*, increased utilization of carbon energy sources, increased metabolism in presence of Krebs cycle intermediates, and increased utilization of amino acids and dipeptides. Reduced utilization of tryptophan was observed, as reported in ASD. Although not deleted, *TNFRSF13C* has reduced expression, which is in line with the TAD prediction; this gene has been associated with schizophrenia (S2 File).

## PMS 4

Decreased expression in *ADSL* and *NAGA* was found for PMS 4 even though they were preserved. *ADSL* is associated with adenylosuccinase deficiency (ADSLD), a disorder marked with psychomotor retardation, epilepsy, and autistic features. *NAGA* is associated with Schindler disease types I and II, and thus could be the etiology underlying cognitive impairment.

Deletion of *SHANK3* could underlie high levels of anxiety and problems with adaptive behavior in this individual (S2 File). *TUBGCP6*, *SBF1*, and *ARSA* are deleted in this individual concordant with levels of expression lower than when preserved, except in *ARSA*, which is inconclusive. The Biolog plate PM-M1 has a similar profile as for PMS 3 and PMS 5, with increased utilization of carbon energy sources in 70 out of 96 wells. The profile for amino acids is more similar to PMS 1 and PMS 2 (large deletion-size). As reported in individuals with ASD, reduced utilization of tryptophan was observed. In PM-M5, 79.2% of the 96 wells showed a reduced energetic response. A similarly reduced energetic response was observed in M6 and M7, with 91.7% and 97.9% respectively. On the other hand, all the wells in M8 showed the same trend.

## PMS 5

Intragenic *SHANK3* deletion is a probable etiology for the observed intellectual disability, disharmonic organization of personality, significant sleep disturbance, seizures, and other symptoms (S1 File). All the preserved genes within the TAD region (*TUBGCP6*, *SBF1*, and *ARSA*) show an elevated expression. The metabolic profile of this case appeared different from the others probably because of the smallest deletion size encompassing only part of *SHANK3*. PM-M1 shows a similar profile to PMS 3 and PMS 4 (mid and small deletion-sizes), with increased utilization of carbon energy sources only in 45 out of 96 wells. There was a general trend of a lower number of wells with abnormal response as the deletion-size decreased; PMS 5 has the lowest number of wells whereas PMS 1 and PMS 2 have the highest numbers. Increased utilization of amino acids, dipeptides, and tryptophan was also observed for this individual.

## Correlation between clinical, genetic, and metabolic features

The major trends emerging from the metabolic profiling were mostly aligned with the deletion sizes, with the exception of PMS 4. No particular metabolic pattern appeared to be shared by all five individuals with PMS, confirming that the variability observed in PMS clinical presentation and genetic etiology is reflected in the metabolic profile at the cellular level. As expected, individuals with the largest deletions (PMS 1 and PMS 2) showed more disrupted pathways and their metabolic profiles indicated an overall decrease in the cells' capacity to generate energy from different sources, an impairment that can severely affect cell types with high energy needs such as neurons, therefore, resulting in abnormalities in neurodevelopment. Interestingly, PMS 5 shows a correlation between genetic and metabolic features, having an intragenic *SHANK3* deletion and the least severe metabolic profile, but not between genetic and clinical features, as he presented with severe neurobehavioral and gastrointestinal issues. This suggests that other genetic factors may contribute to the individual's phenotype even if they do not appear to be able to significantly affect the metabolic profile. In fact, PMS 5 has a duplication on 12p13.2, in addition to the *SHANK3* deletion and, even if there are no known phenotypic traits associated with this duplication, this indicates a complex genomic rearrangement that may involve different pathogenic mechanisms than ones usually observed in PMS.

## Limitations and future studies

The validation portion of the study was performed on five individuals with deletions representative of the spectrum of deletion sizes and demonstrates the confirmation of computational algorithm predictions with qPCR and Biolog. A larger sample size in future studies can provide validation for our findings. qPCR was performed using blood-derived LCLs which was a limitation for the expression analysis of genes not expressed in blood (*FBLN1*, *SHANK3*, and

*SCO2*). The qPCR approach represents only the first level of validation for TAD prediction; next steps that may be considered include high throughput analyses such as RNA sequencing, which would also show an impact on all genes in the target area and not only on target genes.

Overall, the results confirmed clinical and genetic variability within the PMS cohort, with more frequent and severe features associated with larger 22q13 deletions. Such correlation was observed also in the metabolic arrays, where individuals with the largest deletions (PMS 1 and PMS 2) showed clearly distinct profiles, characterized by significant impairment of several energetic pathways. Although metabolomics is becoming more and more important for clinical diagnostics of genetic disorders, and the technique is still in its infancy and reliable data require a number of patients and controls well above those used in this study, the numbers are still significant for a rare disorder. The TAD analysis successfully predicted the disruptive position effects for the genes deleted, except for *A4GALT* as discussed. Other candidate genes for dysregulation by position effect had decreased transcript levels when deleted, suggesting that their loss, regardless of the mechanism, will affect their expression.

Our findings presented in this study support a contributory role of several genes in 22q13, and in some cases suggest some specific correlations with PMS features, such as *SBF1* and *ARSA* for neurological symptoms. The results also suggest that novel strategies of investigation, such as the TAD analysis and the Biolog PM-M arrays, may be instrumental for better characterizing the complex genotype-phenotype correlation in PMS.

## Supporting information

**S1 File. Sheet 1.** 114 PMS-associated deletions positions detected by microarray. **Sheet 2.** hg19. 91 patient deletions. Deletions converted from hg18 to hg19 using the UCSC LiftOver tool [31]. Only 82 out of a total of 91 converted cases had available phenotypic information. **Sheet 3.** Clinical phenotype for 81 patients. **Sheet 4.** TAD analysis for 81 patients. Grey indicates genes that are candidates but fall within the deletion region; yellow indicates position effect candidates that are outside the deletion. **Sheet 5.** qRT-PCR summary of expression. P1-P5 represent the five individuals with PMS chosen for this study, and C1-C2 represent the two controls used for qRT-PCR experiments. **Sheet 6.** p-values for Biolog plates PM-M1, PM-TRP, PM-M5, and PM-M7. p-values for all five individuals with PMS with significant values highlighted in pink. Only wells with significant differences as compared to controls are selectively shown here.
(XLSX)

**S2 File. Notes.** Notes for sheets 1–4 in the S1 File. **Fig 1.** Deletion breakpoints for 91 individuals with PMS used for TAD analysis. **Fig 2.** Parametric analysis of Biolog plates 1–8 for individuals PMS 1 through PMS 5 versus 50 controls. **Table.** Genes selected for this study with their aliases and functions. Information from the Gene database from NCBI website.
(DOCX)

**S3 File.**
(XLSX)

## Acknowledgments

The authors would like to thank Cindy Skinner and Melanie May (Greenwood Genetic Center) for coordinating the collection of biosamples, the families of individuals with PMS and the Phelan-McDermid Syndrome Foundation (PMSF), Canadian PMS Foundation, and Italian Association for Phelan-McDermid Syndrome (AISPHEM ONLUS).

## Author Contributions

**Conceptualization:** Cinthya Zepeda-Mendoza, Barb DuPont, Sara Sarasua, Katy Phelan, Cynthia Morton, Luigi Boccuto.

**Data curation:** Lavanya Jain, Cinthya Zepeda-Mendoza, Rini Pauly, Sara Sarasua, Luigi Boccuto.

**Formal analysis:** Cinthya Zepeda-Mendoza, Rini Pauly, Cynthia Morton.

**Investigation:** Sujata Srikanth, Lavanya Jain, Cinthya Zepeda-Mendoza, Lauren Cascio, Kelly Jones.

**Methodology:** Cinthya Zepeda-Mendoza, Rini Pauly, Barb DuPont.

**Project administration:** Luigi Boccuto.

**Supervision:** Barb DuPont, Curtis Rogers, Sara Sarasua, Katy Phelan, Cynthia Morton, Luigi Boccuto.

**Validation:** Curtis Rogers, Luigi Boccuto.

**Visualization:** Curtis Rogers.

**Writing – original draft:** Sujata Srikanth, Lavanya Jain, Lauren Cascio, Kelly Jones, Sara Sarasua, Luigi Boccuto.

**Writing – review & editing:** Cinthya Zepeda-Mendoza, Rini Pauly, Barb DuPont, Curtis Rogers, Sara Sarasua, Katy Phelan, Cynthia Morton, Luigi Boccuto.

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
