## [Decision Letter · Decision Letter 0]

1 Apr 2021

PONE-D-21-07656

Position effects of 22q13 rearrangements on candidate genes in Phelan-McDermid syndrome.

PLOS ONE

Dear Dr. Boccuto,

Thank you for submitting your manuscript to PLOS ONE. After careful consideration, we feel that it has merit but does not fully meet PLOS ONE’s publication criteria as it currently stands. Therefore, we invite you to submit a revised version of the manuscript that addresses the points raised during the review process.

Reviewers addressed precise and thoughtful comments. The main concern of reviewer 2 is that the sample size for validation of the in silico analysis is an important limitation of this study.

We look forward to receiving your revised manuscript.

Kind regards,

Barbara Bardoni

Academic Editor

PLOS ONE

Journal Requirements:

2. We note that you are reporting an analysis of a microarray, next-generation sequencing, or deep sequencing data set. PLOS requires that authors comply with field-specific standards for preparation, recording, and deposition of data in repositories appropriate to their field. Please upload these data to a stable, public repository (such as ArrayExpress, Gene Expression Omnibus (GEO), DNA Data Bank of Japan (DDBJ), NCBI GenBank, NCBI Sequence Read Archive, or EMBL Nucleotide Sequence Database (ENA)). In your revised cover letter, please provide the relevant accession numbers that may be used to access these data. For a full list of recommended repositories, see http://journals.plos.org/plosone/s/data-availability#loc-omics or http://journals.plos.org/plosone/s/data-availability#loc-sequencing.

"This study was supported in part by the Greenwood Genetic Center Foundation and the Hope for 22q13 Gala for their kind collaboration and continuous support to the project. Additional support was provided by the Clemson Support for Early Exploration and Development (CU SEED) Grant Program."

"The authors received no specific funding for this work."

4. Please ensure that you refer to Figure 4 in your text as, if accepted, production will need this reference to link the reader to the figure.

Additional Editor Comments (if provided):

The two reviewers addressed precise and thoughtful comments and we ask you to answer to them point by point. Importantly, a main concern from reviewer 2 is that the sample size used for validation of the in silico analysis is an important limitation of the study.

Reviewers' comments:

Reviewer's Responses to Questions

**Comments to the Author**

1. Is the manuscript technically sound, and do the data support the conclusions?

Reviewer #1: Yes

Reviewer #2: Partly

2. Has the statistical analysis been performed appropriately and rigorously? 

Reviewer #1: I Don't Know

Reviewer #2: Yes

3. Have the authors made all data underlying the findings in their manuscript fully available?

Reviewer #1: Yes

Reviewer #2: Yes

4. Is the manuscript presented in an intelligible fashion and written in standard English?

Reviewer #1: Yes

Reviewer #2: Yes

5. Review Comments to the Author

Reviewer #1: Overview

The authors address phenotype variability of Phelan-McDermid syndrome (PMS) through topologically associated domain analysis (TAD). Although there is ample evidence for the involvement of genes throughout the ~ 9 Mb region of observed deletions in the PMS population, it is often difficult to reconcile CMA-determined deletion breakpoints and observed phenotypes. Methodologically, this study of potential position effects leverages the 2017 AJHG work by Zepeda-Mendoza et al (their reference #30). A mix of predicted TAD sites, haploinsufficiency scores and available cases produced two lists, a list of candidate genes and a list of cases where positional effects might impact gene expression. An additional analysis expands the phenotype information by including metabolic profiles from patient-derived lymphoblastoid cell lines (LCLs). Not unexpectedly, the results rest on detailed analysis of relatively few clinical cases (validation sample). The authors demonstrate gene expression variability. Given the nature of the wide range of chromosomal deletions in PMS, the authors were able to directly compare gene loss to putative positional effects within the same genetic syndrome. The strengths of this work are that it addresses phenotype variability of a rare neurodevelopmental disorder by including both gene deletions and positional effects, and it integrates metabolomics with other phenotype data. The results strongly support deletion size effects, position effects and the ability of metabolomics to server as a biomarker. The small number of cases limits the study to one of validation. However, the results are intriguing in that they suggest a personalized approach to prognosis may be within reach. This work can serve as a model for other rare diseases characterized by chromosomal anomalies. As such, it is an important contribution.

The manuscript quality is good, but it has technical issues that require attention.

Specific points

Introduction

The Introduction explains the background and rationale well until p6, line 128. “Nine genes were selected from this list…” Was this selected from the supplementary tables of Zepeda-Mendoza et al or was this a new list generated from an independent analysis? Materials and methods indicate the latter. Perhaps the Introduction can be worded more clearly.

Materials and Methods

Cohort of individuals with PMS

There is a detailed description of one cohort and, as best as this reviewer can decipher, no description of the “independent cohort”. It was not clear to what degree equivalent methods were used to recruit participants, identify breakpoints, classify phenotype or even gain written consent. A much more clear description is needed. If the “independent cohort” is to remain anonymous, the authors can achieve an adequate description by identifying key similarities and differences from the primary cohort.

Explicitly indicate the number of males and females among the 5 individuals selected for the study. Indicate the number of controls and their sexes.

Selection of 12 candidate genes

The haploinsufficiency (HI) scores are noted towards the end of p7, line 157. These are weighted against clinical features. In the original work by Zepeda-Mendoza et al it seems that HI scores <10% were required for case consideration (9 of the 16 top ranked genes in that study). In the present PMS study only two of the 8 candidates for long range position effects have HI scores <10% (Results) and it seems the HI scores were used for ranking. If there was a difference in how phenomatch scores and HI were used in the two studies, that difference should be noted or explained.

The value <10% is used in some spots and <=10% is used elsewhere (i.e. p8, line 160). This is minor but consistency would be appreciated.

Finally, what does it mean in the context of this study to consider a gene pathogenic or not? There is very little mention of pathogenicity elsewhere in the manuscript.

Minor: p7, line 154 “degree by which”. Perhaps “degree to which”?

Minor: Figure 4 uses %HI and Figure 3 uses HI%.

Validation sample

See the above comment about the “independent cohort”.

Table 1. Definition of deletion size. While there is no universal definition of a “Large”, “Mid” or “Small” deletion, one wonders how a deletion if 7.44 Mb can be considered Mid when the population gene deletion size average (both median and mean) is below 5 Mb? Likewise, 3.46 Mb is closer to the average than 7.44 Mb and is labeled as Small. In addition, using the classifications later in the manuscript (pp26-27) implies a generalization that is not sufficiently supported by the limited data. The authors are asked to either justify these monikers, omit them, or at least declare them arbitrary. On a related note, the last column in Table 1 (if not omitted) should be more consistent with the rest of the table. For example, each row might include the “size” for that row.

Results

Computational Prediction Algorithm

The first paragraph should reference Fig 4, since it represents the primary findings of the study and is the main supporting evidence for the last sentence in that paragraph. In fact, Figure 4 is not referenced at all in the text. There are two references to Fig 3a that seem out of place (p19, line 388 and p24, line 504). Perhaps these were intended to reference Fig 4.

Table 2 legend. Remove the phrase about “SHANK3”.

Figure 3a. The authors could show the 2 Mb windows for each breakpoint used for TAD analysis. This is a minor point.

Figure 3c. The color scheme makes it very difficult to distinguish PMS 1 from PMS 5. It is strongly recommended that the colors scheme be modified.

Figure 4. The legend and body text should state that the tables in (a) and (b) are exactly the same data displayed with two different shading schemes. Omit the phrase regarding SHANK3.

Clinical features of selected individuals with PMS

The descriptions do not follow a fixed format making the cases somewhat more difficult to compare. The most striking annoyance is not including the deletion size in the first (or second) sentence of the description for PMS 5. Please put the deletion size early in the first paragraph. The details of the less relevant elements of the rearrangement can be described later.

Expression analysis by qPCR for selected candidate genes

As noted before, references to large-size, mid-size and small-size deletions are not necessary or especially helpful. Remove the individual category headings. The section is already broken up with a paragraph for each case and the deletion sizes (nicely indicated for each case) shows the monotonic size decrements.

Discussion

Organizing the discussion by cases PMS 1 through PMS 5 may not be the preferred approach in a paper about the genes affected. Each section refers to other cases and it might be possible to write a more consolidated discussion. That said, the present organization is satisfactory. Note p25, line 527 “whenever ill ocular bilateral red reflex was observed”. Something is amiss.

Limitations and future studies

No changes necessary.

Reviewer #2: This report tested if deletions at 22q13, causative of Phelan McDermid syndrome (PMS), change the expression of neighbour genes by position effect. Based on the use of computational predictive logarithm in 81 PMS individuals, 12 candidate genes were prioritized. Nine of these 12 genes were validated in lymphoblastoid cell lines (LCL) of 5 PMS individuals with different 22q13 terminal deletions. Correlation of gene expression and phenotype was also conducted. Metabolic profile under exposition of different compounds was also done in LCL and controls and correlated with expression and clinical data.

The question addressed is new and worth to be investigated.

Major concern:

Number of validation cases are too small. The number of cases of altered expression of the candidate gene is not replicated in the small validation sample.

Other comments:

1) Page 12,

Line 259: “Except for A4GALT, all candidates exhibited phenotypic overlap with the clinical features of the cases analysed…

Line 262: “Noteworthy, not all candidates were detected in all PMS individuals analyzed. Some of the genes were detected in one individual and other in up to nine individuals”.

These sentences are very difficult to follow up.

2) Four of the 12 genes predicted to be dysregulated by the use of a computational algorithm were due to deletion. Is it the expected number to be detected in this cohort?

3) Most of the observations of altered expression was based on observation on a single individual, which is not strong enough to support the hypothesis of position effect. The variability of these genes in a large number of LCL of controls should be presented.

4) Page 31, line489 : “A4GALT represents an exception to some extent because its expression pattern was not as expected and differs from ones of surrounding genes with increased expression regardless of the deletion size. One possible explanation for this unexpected trend is that it encodes an enzyme and could therefore be regulated by a compensatory mechanism aimed to prevent haploinsufficiency.”

The authors should comment on the function of this gene and better explain the hypothesis of the compensatory mechanisms to prevent haploinsufficiency. Notably, ADSL, also validated in the expression experiments, is also an enzyme.

5) Except of EP300, all the other OMIM genes are associated with autosomal recessive disorders. How does loss-of-function of one allele in these genes could contribute to phenotype?

6) Most of the phenotype-correlation presented in this report was based in single patients, which are not sufficient to support the position effect hypothesis

7) I did not find details of the controls listed in Sheet 4 /S1 mentioned in methods.

6. PLOS authors have the option to publish the peer review history of their article (what does this mean?). If published, this will include your full peer review and any attached files.

Reviewer #1: **Yes: **Andrew R. Mitz

Reviewer #2: No

---

## [Author Response · Author response to Decision Letter 0]

19 May 2021

To the Editor in Chief,

PLOS ONE 

We would like to thank the Reviewers for the time they dedicated to our manuscript and for their insightful comments. We have addressed their requests in the main text and provided a point-by-point response below.

Reviewer #1: Overview

The authors address phenotype variability of Phelan-McDermid syndrome (PMS) through topologically associated domain analysis (TAD). Although there is ample evidence for the involvement of genes throughout the ~ 9 Mb region of observed deletions in the PMS population, it is often difficult to reconcile CMA-determined deletion breakpoints and observed phenotypes. Methodologically, this study of potential position effects leverages the 2017 AJHG work by Zepeda-Mendoza et al (their reference #30). A mix of predicted TAD sites, haploinsufficiency scores and available cases produced two lists, a list of candidate genes and a list of cases where positional effects might impact gene expression. An additional analysis expands the phenotype information by including metabolic profiles from patient-derived lymphoblastoid cell lines (LCLs). Not unexpectedly, the results rest on detailed analysis of relatively few clinical cases (validation sample). The authors demonstrate gene expression variability. Given the nature of the wide range of chromosomal deletions in PMS, the authors were able to directly compare gene loss to putative positional effects within the same genetic syndrome. The strengths of this work are that it addresses phenotype variability of a rare neurodevelopmental disorder by including both gene deletions and positional effects, and it integrates metabolomics with other phenotype data. The results strongly support deletion size effects, position effects and the ability of metabolomics to server as a biomarker. The small number of cases limits the study to one of validation. However, the results are intriguing in that they suggest a personalized approach to prognosis may be within reach. This work can serve as a model for other rare diseases characterized by chromosomal anomalies. As such, it is an important contribution.

The manuscript quality is good, but it has technical issues that require attention.

Specific points

Introduction

The Introduction explains the background and rationale well until p6, line 128. “Nine genes were selected from this list…” Was this selected from the supplementary tables of Zepeda-Mendoza et al or was this a new list generated from an independent analysis? Materials and methods indicate the latter. Perhaps the Introduction can be worded more clearly.

We thank the Reviewer for this observation. The paragraph has been modified to clarify that the analysis was performed in our PMS deletion cohort using Zepeda-Mendoza's et al algorithm. The nine candidate genes analyzed in our manuscript are from our candidate genes list, not that of Zepeda-Mendoza et al.

Materials and Methods

Cohort of individuals with PMS

There is a detailed description of one cohort and, as best as this Reviewer can decipher, no description of the “independent cohort”. It was not clear to what degree equivalent methods were used to recruit participants, identify breakpoints, classify phenotype or even gain written consent. A much more clear description is needed. If the “independent cohort” is to remain anonymous, the authors can achieve an adequate description by identifying key similarities and differences from the primary cohort.

- We apologize for the lack of clarity: the independent cohort was merely constituted by the two additional subjects. In order to avoid further confusion, we replaced “independent cohort” with “unrelated subjects/patients” in the manuscript.

Explicitly indicate the number of males and females among the 5 individuals selected for the study. 

- The number of males and females (four males and one female) is indicated on page 8, line 164, we have added such clarification also on page 9, line 181.

Indicate the number of controls and their sexes.

- We have specified the sex of the two controls used for the expression study (page 9, line 181) and we added a citation about the 50 controls used for the Phenotype Mammalian MicroArray experiments on page 11, lines 243-244: “previously utilized in a study on ASD [33]”.

Selection of 12 candidate genes

The haploinsufficiency (HI) scores are noted towards the end of p7, line 157. These are weighted against clinical features. In the original work by Zepeda-Mendoza et al it seems that HI scores <10% were required for case consideration (9 of the 16 top ranked genes in that study). In the present PMS study only two of the 8 candidates for long range position effects have HI scores <10% (Results) and it seems the HI scores were used for ranking. If there was a difference in how phenomatch scores and HI were used in the two studies, that difference should be noted or explained.

- We thank the Reviewer for raising this very relevant concern regarding the analysis methodology. HI scores <10% are one of the four criteria used for the overall ranking of the candidates, the other three being disruption of known or predicted regulatory contacts, location within the 2 Mb/TAD windows, and phenotypic overlap above the 75th percentile. So while a candidate gene might not be happloinsufficient, it might have hit the other three requirements. We have clarified this in the analysis segment, which now includes in line 159 " Position effect candidates were ranked given their disruption of known or predicted regulatory contacts, location within the 2 Mb/TAD windows, HI scores (=<10%), and phenotypic overlap above the 75th percentile [30]."

The value <10% is used in some spots and <=10% is used elsewhere (i.e. p8, line 160). This is minor but consistency would be appreciated.

- All instances of <10% were edited to =<10%.

Finally, what does it mean in the context of this study to consider a gene pathogenic or not? There is very little mention of pathogenicity elsewhere in the manuscript.

In terms of this study, a brief definition for gene pathogenicity in PMS can be found in line 125 of the introduction: " These genes were selected based on their roles in brain activity, expression in whole blood, and their location on the 22q13.3 genomic region." In some cases, variants are known to be pathogenic, such as selected SHANK3 variants, whereas in other cases pathogenicity can only now be proposed until additional work identifies specific mechanisms.

Minor: p7, line 154 “degree by which”. Perhaps “degree to which”?

- We edited the text as suggested.

Minor: Figure 4 uses %HI and Figure 3 uses HI%.

- We replaced HI% and used %HI throughout the text.

Validation sample

See the above comment about the “independent cohort”.

- We have removed “independent cohort” and used “unrelated subjects/patients”.

Table 1. Definition of deletion size. While there is no universal definition of a “Large”, “Mid” or “Small” deletion, one wonders how a deletion if 7.44 Mb can be considered Mid when the population gene deletion size average (both median and mean) is below 5 Mb? Likewise, 3.46 Mb is closer to the average than 7.44 Mb and is labeled as Small. In addition, using the classifications later in the manuscript (pp26-27) implies a generalization that is not sufficiently supported by the limited data. The authors are asked to either justify these monikers, omit them, or at least declare them arbitrary. On a related note, the last column in Table 1 (if not omitted) should be more consistent with the rest of the table. For example, each row might include the “size” for that row.

- We agree with the Reviewer and clarified our classification in the notes of Table 1: “Such definition is arbitrary and purely based on the distribution of the genes in the 22q13 region. The sole purpose of the distinction of the deletion size into three categories is to provide a rough stratification of the contribution of different loci on 22q13.” We have also edited the last column of Table 1 as suggested.

Results

Computational Prediction Algorithm

The first paragraph should reference Fig 4, since it represents the primary findings of the study and is the main supporting evidence for the last sentence in that paragraph. In fact, Figure 4 is not referenced at all in the text. There are two references to Fig 3a that seem out of place (p19, line 388 and p24, line 504). Perhaps these were intended to reference Fig 4.

- We added references to Figure 4 as suggested on the first paragraph of the Computational Prediction Algorithm section and on page 19. We believe that the reference to Figure 3a on page 24 is correct since it refers to the %HI scores indicated in green next to the gene names in the figure.

Table 2 legend. Remove the phrase about “SHANK3”.

- We removed the phrase as suggested.

Figure 3a. The authors could show the 2 Mb windows for each breakpoint used for TAD analysis. This is a minor point.

 - We added a 2 Mb blue window adjoining the starting breakpoint in Figure 3a for each individual.

Figure 3c. The color scheme makes it very difficult to distinguish PMS 1 from PMS 5. It is strongly recommended that the colors scheme be modified.

 - We changed the color for PMS 5 in Figure 3c.

Figure 4. The legend and body text should state that the tables in (a) and (b) are exactly the same data displayed with two different shading schemes. Omit the phrase regarding SHANK3.

- This statement was added to the legend for Figure 4 as well as in the body text on page 13. 

Clinical features of selected individuals with PMS

The descriptions do not follow a fixed format making the cases somewhat more difficult to compare. The most striking annoyance is not including the deletion size in the first (or second) sentence of the description for PMS 5. Please put the deletion size early in the first paragraph. The details of the less relevant elements of the rearrangement can be described later.

- We added the genetic information to PMS 5 as suggested: “carrying a 100.76 Kb 22q13 deletion encompassing the SHANK3 gene detected by array-CGH”.

Expression analysis by qPCR for selected candidate genes

As noted before, references to large-size, mid-size and small-size deletions are not necessary or especially helpful. Remove the individual category headings. The section is already broken up with a paragraph for each case and the deletion sizes (nicely indicated for each case) shows the monotonic size decrements.

- We removed the headings as suggested.

Discussion

Organizing the discussion by cases PMS 1 through PMS 5 may not be the preferred approach in a paper about the genes affected. Each section refers to other cases and it might be possible to write a more consolidated discussion. That said, the present organization is satisfactory. Note p25, line 527 “whenever ill ocular bilateral red reflex was observed”. Something is amiss.

- We agree with the Reviewer about the organization of the discussion: our strategy revolved on the purpose of correlating genetic, clinical, and metabolic features for each patient, aiming to highlight the common trends across the tested cohort. We adopted this organization because we believed that the number of patients for which all analyses have been performed was too limited to allow a more consolidated discussion. We edited line 527 of page 25 into “ocular bilateral red reflex that was observed whenever the patient was ill.”

Limitations and future studies

No changes necessary.

Reviewer #2: This report tested if deletions at 22q13, causative of Phelan McDermid syndrome (PMS), change the expression of neighbour genes by position effect. Based on the use of computational predictive logarithm in 81 PMS individuals, 12 candidate genes were prioritized. Nine of these 12 genes were validated in lymphoblastoid cell lines (LCL) of 5 PMS individuals with different 22q13 terminal deletions. Correlation of gene expression and phenotype was also conducted. Metabolic profile under exposition of different compounds was also done in LCL and controls and correlated with expression and clinical data.

The question addressed is new and worth to be investigated.

Major concern:

Number of validation cases are too small. The number of cases of altered expression of the candidate gene is not replicated in the small validation sample.

 - We understand and agree with the Reviewer’s concern. We are aware that the number of cases in the validation cohort is small, but our main goal was to generate a pilot study that could provide functional evidence in support of the probabilistic scenarios generated by the TAD predictions in a model disorder such as PMS. We recognize that future studies, performed on larger cohorts are necessary to validate the trends we reported in our work.

Other comments:

1) Page 12,

Line 259: “Except for A4GALT, all candidates exhibited phenotypic overlap with the clinical features of the cases analysed…

Line 262: “Noteworthy, not all candidates were detected in all PMS individuals analyzed. Some of the genes were detected in one individual and other in up to nine individuals”.

These sentences are very difficult to follow up.

- We replaced the first sentence with “Except for A4GALT, all candidate genes have been associated with phenotypic traits compatible with the clinical features reported in the cases analyzed.” We replaced the second sentence with “Noteworthy, not all candidate genes were detected in every individual with PMS analyzed.”

2) Four of the 12 genes predicted to be dysregulated by the use of a computational algorithm were due to deletion. Is it the expected number to be detected in this cohort?

- Unfortunately there is no expected number of position effect candidates to be obtained per any given genomic region, since gene density, chromatin organization, and abundance of regulatory elements is so varied across the genome. 

3) Most of the observations of altered expression was based on observation on a single individual, which is not strong enough to support the hypothesis of position effect. The variability of these genes in a large number of LCL of controls should be presented.

- We agree with the Reviewer, but unfortunately we are not able to perform further expression studies on control LCLs at the moment: the project was only meant to be a pilot study and most of the investigators have no longer access to the original samples or new ones.

4) Page 31, line489 : “A4GALT represents an exception to some extent because its expression pattern was not as expected and differs from ones of surrounding genes with increased expression regardless of the deletion size. One possible explanation for this unexpected trend is that it encodes an enzyme and could therefore be regulated by a compensatory mechanism aimed to prevent haploinsufficiency.”

The authors should comment on the function of this gene and better explain the hypothesis of the compensatory mechanisms to prevent haploinsufficiency. Notably, ADSL, also validated in the expression experiments, is also an enzyme.

- We added the following sentence: “in case one allele was either lost by deletion or inactivated, the remaining allele would be over-expressed to maintain a physiological level of protein.” Further details about the gene function are provided in the Supplemental table. 

5) Except of EP300, all the other OMIM genes are associated with autosomal recessive disorders. How does loss-of-function of one allele in these genes could contribute to phenotype?

- We believe that loss or inactivation of one allele may unmask heterozygous variants on the remaining active allele.

6) Most of the phenotype-correlation presented in this report was based in single patients, which are not sufficient to support the position effect hypothesis

- We are aware and agree with the Reviewer: this is a pilot study that aimed to apply multiple approaches to the assessment of the contribution of various 22q13 genes to the phenotype of PMS. We concur that further evidence in support of the trends reported in this manuscript needs to be collected on larger cohorts.

7) I did not find details of the controls listed in Sheet 4 /S1 mentioned in methods.

- We added a citation about the 50 controls used for the Phenotype Mammalian MicroArray experiments on page 11, lines 243-244: “previously utilized in a study on ASD [33]”.

Sincerely,

Luigi Boccuto

---

## [Decision Letter · Decision Letter 1]

15 Jun 2021

Position effects of 22q13 rearrangements on candidate genes in Phelan-McDermid syndrome.

PONE-D-21-07656R1

Dear Dr. Boccuto,

We’re pleased to inform you that your manuscript has been judged scientifically suitable for publication and will be formally accepted for publication once it meets all outstanding technical requirements.

Kind regards,

Barbara Bardoni

Academic Editor

PLOS ONE

Additional Editor Comments (optional):

Reviewers' comments:

Reviewer's Responses to Questions

**Comments to the Author**

1. If the authors have adequately addressed your comments raised in a previous round of review and you feel that this manuscript is now acceptable for publication, you may indicate that here to bypass the “Comments to the Author” section, enter your conflict of interest statement in the “Confidential to Editor” section, and submit your "Accept" recommendation.

Reviewer #1: All comments have been addressed

Reviewer #2: (No Response)

2. Is the manuscript technically sound, and do the data support the conclusions?

Reviewer #1: (No Response)

Reviewer #2: Yes

3. Has the statistical analysis been performed appropriately and rigorously? 

Reviewer #1: (No Response)

Reviewer #2: Yes

4. Have the authors made all data underlying the findings in their manuscript fully available?

Reviewer #1: (No Response)

Reviewer #2: Yes

5. Is the manuscript presented in an intelligible fashion and written in standard English?

Reviewer #1: (No Response)

Reviewer #2: Yes

6. Review Comments to the Author

Reviewer #1: (No Response)

Reviewer #2: This revised manuscript has incorporated all the main reviewers' comments.The paper brings new insights to understand clinical variability of patients with 22q13 deletions and open a new perspective to address this issue.

7. PLOS authors have the option to publish the peer review history of their article (what does this mean?). If published, this will include your full peer review and any attached files.

Reviewer #1: **Yes: **Andrew R. Mitz

Reviewer #2: No

---

## [Editor Report · Acceptance letter]

23 Jun 2021

PONE-D-21-07656R1 

Position effects of 22q13 rearrangements on candidate genes in Phelan-McDermid syndrome 

Dear Dr. Boccuto:

I'm pleased to inform you that your manuscript has been deemed suitable for publication in PLOS ONE. Congratulations! Your manuscript is now with our production department. 

Kind regards, 

on behalf of

Dr. Barbara Bardoni 

Academic Editor

PLOS ONE